# The Effect of Two-Generation Exposure to a High-Fat Diet on Craniofacial Morphology in Rats

**DOI:** 10.3390/jcm12051903

**Published:** 2023-02-28

**Authors:** Saranya Serirukchutarungsee, Ippei Watari, Pornchanok Sangsuriyothai, Masato Akakura, Takashi Ono

**Affiliations:** 1Department of Orthodontic Science, Graduate School of Medical and Dental Sciences, Tokyo Medical and Dental University (TMDU), Tokyo 113-8510, Japan; 2Department of Pedodontics and Preventive Dentistry, Faculty of Dentistry, Srinakharinwirot University, Bangkok 10110, Thailand; 3Department of Orthodontics, Faculty of Dentistry, Chulalongkorn University, Bangkok 10330, Thailand

**Keywords:** high-fat diet, pregnancy, lactation, craniofacial growth, cephalometric analysis

## Abstract

This study aimed to examine the sexual dimorphism effect of two-generation exposure to a high-fat diet (HFD) on the craniofacial growth of rat offspring. Ten eleven-week-old pregnant Wistar rats were fed either a control or HFD from day 7 of pregnancy until the end of lactation. Twelve male and female offspring from the control-diet-fed mothers were assigned to the CM (control male, n = 6) and CF (control female, n = 6) groups. The other twelve from the HFD-fed mothers were assigned to the HFD male (HFDM, n = 6) and HFD female (HFDF, n = 6) groups. HFDM and HFDF rats continued with an HFD. The offspring’s weight and fasting blood sugar levels were measured every two weeks. The craniofacial and dental morphologies were studied from lateral X-rays of the head at ten weeks old. The HFDM rats showed an increased body weight and larger neurocranial parameters compared with the CM group. Furthermore, there were slightly significant differences in body weight and viscerocranial parameters between the rats in the HFDF and CF groups. In conclusion, two-generational exposure to an HFD had a greater effect on the male offspring’s body weight and craniofacial morphology.

## 1. Introduction

Dietary patterns and physical activity are considered to be important factors for weight gain [1], where an imbalance between energy intake and expenditure results in excessive fat accumulation [1]. Genetics is an underlying factor that controls hunger, satiety, and behaviors. Relevant gene defects, such as melanocortin 4 receptor (MC4R), Src-homology-2 (SH2B1), and potassium channel tetramerization domain-containing 15 (KCTD15), were found to be associated with excessive eating behavior and overnutrition. Moreover, the obesogenic environment aggravates the adverse effects of an HFD [2,3]. A post-weaning HFD combined with a maternal HFD resulted in hyperphagia and obesity in offspring [2,3,4]. The body weight and adipose tissue size of offspring following two-generation exposure to an HFD were remarkably higher than offspring from either control-fed or one-generational exposure to HFD rats [2,3,4]. These results suggested that post-weaning overnutrition potentially magnified the detrimental effects of an HFD during pregnancy and lactation. Moreover, maternal food patterns tend to induce the same food patterns in offspring [5,6]. Therefore, a post-weaning obesogenic environment is most likely to happen in most situations. Thus, researchers need to recognize the effect of a high-fat diet (HFD), not only during prenatal life but also post-weaning life, which affects the offspring’s long-term health.

A maternal HFD influences bone development in both humans and rodents [7]. A study of 53,922 pairs of mothers and children found an interaction between the maternal intake of a high-fat Western-style diet and forearm fracture in offspring [8]. In rodent studies, the fetal offspring of dams fed a maternal HFD presented decreased bone formation, density, and volume during late gestation [7,9]. Higher senescence in fetal calvarial osteoblast cells potentially interferes with bone formation in fetuses. In contrast, weaned offspring from HFD-fed mothers showed an increased bone mass and density in their long bones [7,10]. An HFD after weaning further reduced trabecular bone volume compared with offspring from HFD-fed dams weaned on the control diet. These results suggest that nutrition is essential to long-term skeletal remodeling in offspring. Moreover, the effect of a maternal HFD could be considerably amplified by a post-weaning HFD, influencing multiple generations [4,7,11,12]. 

Sexual dimorphism of the craniofacial region has been studied for decades [13,14,15]. The differences in anterior cranial base length, mid-facial length, mandibular length, and the face height of males and females were reported [13]. Furthermore, previous studies demonstrated the sexual differences in metabolic response to an HFD [16,17,18]. In rodents, males were more susceptible to body weight gain than females [16,17,18]. However, the sexual dimorphism effects of an HFD on craniofacial morphology have rarely been studied, especially the combination of pre- and post-weaning HFD consumption. Therefore, this study focused on the sexually different effects of a maternal HFD during pregnancy and lactation combined with post-weaning HFD on offspring’s craniofacial morphology using cephalometric analyses. Understanding the correlations between HFD exposure and the development of offspring’s craniofacial patterns will increase the awareness of health professionals toward its adverse effects, improving their ability to construct appropriate treatment plans for patients.

## 2. Materials and Methods

### 2.1. Animals and Experimental Design

Before the study, all animal and experimental procedures were approved by the Institutional Animal Care and Welfare Committee (A2020-148A). They were performed following the Animal Care Standards of Tokyo Medical and Dental University (TMDU) and ARRIVE guidelines.

Ten eleven-week-old Wistar rats were purchased from the Sankyo Labo Service Corporation (Tokyo, Japan). All rats were fed either a control diet (CE2, Clea, Tokyo, Japan; 4.6% from fat, 3.402 kcal/g) or an HFD (HFD32, Clea, Tokyo, Japan; 32% from fat, 5.076 kcal/g) from day 7 of pregnancy until the end of the lactation period. Because the embryonic neural folds develop at seven days of gestation, the diet intervention of this study was started synchronously [19]. The offspring from litters containing nine to twelve pups were randomly selected to standardize the litter sizes [2]. The outsized litters from the study were excluded. Therefore, the conditions during the gestation and lactation of all samples were comparable. One to two pups from each litter were randomly selected. In total, 24 offspring were assigned to 4 groups. The male and female offspring of a mother that consumed the control diet were named the CM (n = 6) and CF (n = 6) groups, respectively. The other male and female offspring (12 each) of the HFD-fed mothers were assigned to the HFDM (n = 6) and HFDF (n = 6) groups, respectively. After weaning, all pups continued consuming the same diet as their mothers to mimic an obesogenic environment (Figure 1).

### 2.2. Body Weight and Fasting Blood Glucose Measurement

Body weight was measured every 2 weeks from weaning until the end of the experiment at the age of 10 weeks old. An HFD compromised beta cell development and function and contributed to the development of obesity and insulin resistance [20]. Moreover, high blood sugar levels resulted in craniofacial morphology alteration [21,22]. Therefore, the fasting blood sugar level was observed during the growth period in this study. All rats fasted for 8 h before the measurement of fasting blood sugar (FBS) levels at 4, 6, 8, and 10 weeks old.

### 2.3. Cephalometric Analyses

At 10 weeks old, the rats were anesthetized using the three types of mixed anesthesia prepared with medetomidine hydrochloride 0.375 mg/kg (Orion, Hokkaido, Japan), midazolam 2 mg/kg (Sandoz, Basel, Switzerland), and butorphanol tartrate 2.5 mg/kg (Meiji Seika, Tokyo, Japan) [23]. Each rat’s head was fixed firmly with ear rods, with plastic rings for incisors. Then, lateral cephalometric radiographs were taken using a soft X-ray machine (SOFTEXCMB-2, SOFTEX Co., Ltd., Tokyo, Japan) at 50 kVp, 15 mA, and 20 s exposure. The films (Fujifilm, Tokyo, Japan) were developed and scanned at a high resolution (400 dpi, 16 bit) in .tif format. The twenty-two cephalometric landmarks (Table 1) and twenty-eight linear measurements (Table 2 and Table 3) were obtained from previous studies [21,24,25]. All landmarks and linear measurements were analyzed twice using ImageJ software (Wayne Rasband, NIH, Maryland, USA) to ensure reliability and replicability (Figure 2 and Figure 3). Finally, Dahlberg’s formula was used to determine the reproducibility of the measurements and the method error [26].

### 2.4. Statistical Analysis

The power analysis was done using G*Power version 3.1.9.6. The effect size was calculated from the data of the pilot study. Then, the sample size (n = 6) was determined. All statistical analyses were performed using GraphPad Prism 9 (GraphPad version 9.4.0, USA). The normal distribution of the data was checked using the Shapiro–Wilk test. The statistical significance was set at *p* < 0.05. The data are presented as the mean ± standard deviation (SD). The body weight and FBS data were determined using a one-way ANOVA followed by Tukey’s multiple comparisons test. In addition, all cephalometric analyses were analyzed using a two-way analysis of variance (ANOVA) (the two factors were food and sex) followed by Tukey’s multiple comparisons test. The dental measurement data were not normally distributed. Therefore, all data were analyzed using the Kruskal–Wallis test. The level of significance was set at *p* ≤ 0.05.

## 3. Results

### 3.1. Changes in Body Weight but Not FBS in Offspring

The body weight and FBS of all groups were compared. The body weights of the HFDM and HFDF were significantly higher than those of the CM and CF at 3 weeks old. The HFDM remained heavier than the CM until the end of the experiment, while the body weight of the HFDF was not significantly different from the CF after 3 weeks old. The FBS was not significantly different between all groups throughout the experiment (Figure 4).

### 3.2. Changes in the Cephalometric Parameters 

The method errors (mm) evaluated using the Dahlberg formula were as follows: total skull length (Po–N) = 0.33909, cranial vault length (Po–E) = 0.5606, total cranial base length (Ba–E) = 0.3923, anterior cranial base length (So–E) = 0.3703, occipital bone length (Ba–CB1) = 0.1403, sphenoid bone length (CB1′–CB2) = 0.1448, posterior cranial base length (Ba–So) = 0.2103, posterior neurocranium height (Po–Ba) = 0.4928, nasal length (E–N) = 0.4875, palate length (Mu2–Iu) = 0.2059, midface length (CB2–Iu) = 0.7432, viscerocranial height (E–Mu1) = 0.1447, posterior corpus length (Go–Mn) = 0.9095, anterior corpus length (M1–Il) = 0.2027, total mandibular length (Co–Il) = 1.0589, ramus height (Co–Gn) = 0.6127, maxillary first molar crown width (UM1) = 0.1474, maxillary second molar crown width (UM2) = 0.1258, maxillary third molar crown width (UM3) = 0.1647, mandibular first molar crown width (LM1) = 0.1927, mandibular second molar crown width (LM2) = 0.1934, mandibular third molar width (LM3) = 0.2500, maxillary incisor width (Uiw) = 0.3123, mandibular incisor width (Liw) = 0.9436, maxillary incisor length (Uil) = 0.3327, mandibular incisor length (Lil) = 0.5544, maxillary first molar crown height (UCH) = 0.2919, maxillary first molar root length (URH) = 0.2401, mandibular first molar crown height (LCH) = 0.2035, and mandibular first molar root length (LRH) = 0.2048. 

#### 3.2.1. Changes in the Neurocranium

A two-way ANOVA was conducted to examine the effect of food and sex on all cephalometric parameters. 

In the neurocranium, there was a statistically significant interaction between the effects of food and sex on the total skull length (Po–N), cranial vault length (Po–E), total cranial base length (Ba–E), anterior cranial base length (So–E), occipital bone length (Ba–CB1), and posterior cranial base length (Ba–So).

A simple main effects analysis showed that food had a statistically significant effect on all parameters, with the exception of the anterior cranial base length (So–E), sphenoid bone length (CB1′–CB2), and posterior neurocranium height (Po–Ba), whereas sex affected all parameters, except for the occipital bone length (Ba–CB1) and sphenoid bone length (CB1′–CB2) (Table 4).

The data were then compared between groups using the multiple comparisons test. For the male offspring, the animals in the HFDM group showed longer lengths for all variables, except for the sphenoid bone length (CB1′–CB2) and posterior neurocranium height (Po–Ba) (CM vs. HFDM; Po–N: *p* = 0.009; Po–E: *p* = 0.021; Ba–E: *p* < 0.001; So–E: *p* = 0.038; Ba–CB1: *p* = 0.025; Ba–So: *p* < 0.001). In contrast, there was no significant difference in any of the variables between the CF and HFDF groups (Figure 5A).

#### 3.2.2. Changes in the Viscerocranium

In the viscerocranium, there was a statistically significant interaction between the effects of food and sex on the palate length (Mu2–Iu). A simple main effects analysis showed that food had a statistically significant effect on the palate length (Mu2–Iu) and midface length (CB2–Iu), whereas sex affected all parameters (Table 4).

The palate length (Mu2–Iu) was the only parameter among all of the cephalometric measurements that significantly increased in both the males and females. The midface length (CB2–Iu) was longer in the HFDM group compared with the CM group but not in the HFDF group compared with the CF group (Figure 5B).

#### 3.2.3. Changes in the Mandible

In the mandible, the interaction between food and sex did not affect any parameters. The simple main effects of food statistically affected the anterior corpus length (M1–Il), while sex affected all parameters in the mandible (Table 4). There was no significant difference in all variables among the CM, CF, HFDM, and HFDF groups (Figure 5C).

#### 3.2.4. Changes in the Dental Morphology

It was shown that the HFD did not affect any dental parameters. There were no significant differences between the CM, CF, HFDM, and HFDF groups for every parameter (Figure 6).

## 4. Discussion

Balanced nutrition is essential for growth and development. This study investigated the effects of two-generation exposure to an HFD on the craniofacial growth and morphology of male and female rat offspring using cephalometric analyses. We found the effects of an HFD on craniofacial growth in the different patterns between male and female offspring. At the weaning age, the body weights of the rats in the HFDM and HFDF groups were heavier than their respective CM and CF counterparts. Nonetheless, the body weight of the rats in the HFDM group was continuously larger than that of the CM group throughout the experimental period. In contrast, the body weight of the rats in the HFDF group was not considerably different from the CF group. An HFD similarly affected the males’ neurocranial length more compared with females. The cephalometric analysis showed that the total skull length (Po–N), cranial vault (Po–E), total cranial base (Ba–E), anterior cranial base (So–E), occipital bone (Ba–CB1), and posterior cranial base (Ba–So) lengths of the rats in the HFDM group were significantly longer compared with the CM group. On the other hand, there was no significant difference between any neurocranial parameters for the rats in the HFDF and CF groups. Moreover, the HFD moderately affected the male offspring’s viscerocranium, while it only slightly affected the females. The midface (CB2–Iu) and palate lengths (Mu2–Iu) of the rats in the HFDM group were longer than in the CM group. Interestingly, the palate length (Mu2–Iu) of the rats in the HFDF group was similarly increased compared with those in the CF group. Finally, the effect of an HFD was not detected in any mandible or dental parameters.

The results from our study emphasized the sexual dimorphism of an HFD on craniofacial morphology, which could be the result of sex hormones that influence bone growth and metabolism [27]. Androgens and estrogen enhance craniofacial bone growth, which is evident immediately after birth [27]. The effects of an HFD in this study prominently affected male offspring at ten weeks. The higher intake of fats and energy possibly increases lipid stores and stimulates rapid bone growth [7]. In addition, male hormones can massively stimulate bone extension [27]. As a result, males and females have different skeletal growth patterns [28]. Furthermore, estrogen has a protective effect against an HFD in females by reducing proinflammatory cytokines and maintaining insulin sensitivity during obesity [29]. Therefore, we hypothesized that the different characteristics of male and female sex hormones explained our findings. However, in this study, the mandible was not affected by an HFD in males or females. Mandibular growths are affected by various environmental factors, including postnatal masticatory functions [30]. Our results suggested that the environmental factors were more predominant compared with the effect of an HFD during mandibular development. 

It is important to highlight that a maternal HFD on offspring growth and development could be considerably amplified by a post-weaning HFD, influencing multiple generations [4,7,9,11,12]. Family environment is an important factor associated with obesity in children and adolescents [6]. Obese parents significantly increase the risk of obesity in children [3,6]. Therefore, after weaning, we challenged the offspring with an HFD to mimic the obesogenic environment, which mostly happened in the family. In the current study, the effect of two-generation exposure to an HFD was more considerable for the male than the female offspring. The body weight of the HFDM group was significantly increased compared with those in the CM group throughout the experiment. On the other hand, the body weight of rats in the HFDF group was larger than those in the CF group only at three weeks old. Previous studies reported that a two-generational HFD increased the body weight and post-weaning growth rates of both male and female offspring [2,4,12]. These results emphasized that both genetic and environmental factors played an essential role in offspring health. 

This embryonic craniofacial development involves several transformations and migrations of the ectoderm, mesoderm, and endoderm germ layers [31]. This complex could be affected by maternal conditions and environments because they start from an early stage after fertilization [32,33]. In brief, the embryonic ectoderm changes at the beginning of embryo development and finally forms the neural crest [32]. The neural crest is subsequently differentiated into pharyngeal arches and develops into most of the skull’s bones and cartilages [31,34]. Indeed, most anterior cranial neural crest cells develop into the frontal and nasal bones. In contrast, posterior cranial neural crest cells turn into pharyngeal arches and transform into the maxilla, mandible, middle ear, and a part of the neck [31]. In this study, an HFD appeared to affect each part of craniofacial morphology differently. In males, an HFD had a major effect on the neurocranium. The total skull (Po–N), cranial vault (Po–E), total cranial base (Ba–E), anterior cranial base (So–E), occipital bone (Ba–CB1), and posterior cranial base (Ba–So) lengths of rats in the HFDM group were longer than in the CM group. The minor effect of an HFD was shown in males’ viscerocranium. The midface (CB2–Iu) and palate lengths (Mu2–Iu) of rats in the HFDM group were longer than those in the CM group, while only the palate length (Mu2–Iu) was increased in rats in the HFDF group compared with those in CF. Finally, an HFD did not affect the mandibles and teeth in males or females. The different origins of each part of the skull presumably explain the different effects of an HFD on the neurocranium, viscerocranium, mandibles, and teeth. 

The increased neurocranial growth in the HFDM rats could be the combined effect of maternal and post-weaning HFD. First, maternal HFD was reported to affect the offspring’s bone and craniofacial growth [7,35]. Previous papers explained two possible underlying mechanisms [36,37,38,39,40]. An HFD during pregnancy alters placental gene expression and DNA methylation, increasing the transfer of nutrients across the placenta [36,38,40]. Moreover, the unbalanced maternal lipids during gestation and lactation impair fetal cells. Excessive fat intake during a specific period of pregnancy, as well as lactation, contributes to abnormal cell function and fat metabolism and increases the risk of metabolic diseases in offspring [36]. Second, a post-weaning HFD influenced craniofacial growth in offspring. Previous studies reported long-term consumption of HFD-induced craniofacial morphology changes in rodents [2,12,25,41]. D. Botero-González et al. demonstrated that a post-weaning obesogenic diet decreased the nasal and maxillary lengths in young adult rats [25]. In our study, an HFD did not affect the offspring’s nasal length. The palatal length of rats in the HFDM and HFDF groups increased compared with those in the CM and CF groups. Different types of obesogenic diets could be the reason for the different results. Furthermore, previous studies addressed craniofacial abnormalities in obese patients [41]. Correspondingly with our results, maxillary length increases in obese male and female adolescents [42]. Obesity potentially stimulated growth activity in both genders, resulting in larger craniofacial dimensions. Consequently, various craniofacial characteristics tend to be different in obese patients. In contrast to our study, obese patients tend to have a shorter upper face height, larger mandibles, and flatter and more concave profiles [41]. 

An HFD was reported to influence bones in offspring [7,9,43,44]. An HFD during pregnancy delays fetal skeletal development by suppressing fetal osteoblasts, resulting in decreased bone formation, volume, and mineral density at late gestation [7,9]. Contrastingly, offspring from HFD-fed dams had an increase in bone volume and osteoblast activity after weaning [10]. Higher fat contents in milk were suspected to be the fuel for bone development in offspring [10]. Corresponding to our findings, we speculate that the higher fat content in milk from dams fed an HFD increased the craniofacial growth in male offspring. 

Dental anthropology is determined by multifactorial effects, such as the genetic and environmental factors associated with the development of tooth size [45]. However, the most substantial effect that determines dental size is a specific gene [45,46,47]. In the current study, there was no statistical difference in tooth size between groups. These findings suggest that HFD consumption did not alter the gene sequence. However, a previous study reported that an HFD increased the labial groove and dentine thickness in aged rodents [46,48]. Therefore, detecting the effect of an HFD on teeth probably requires a longer study period and further analysis. 

One of the limitations of this study was the difficulty in separating the direct effect of a maternal HFD on offspring from other environmental and genetic factors. We must consider that our findings were from the effects of mother and offspring. This limitation encourages further studies to clarify these combined effects and their mechanisms. Furthermore, future studies can investigate serum lipid and associated factors coupled with body weight and blood sugar levels. This will allow for an understanding of the mechanisms of our results. Understanding the different patterns of craniofacial growth and the development of obese individuals will increase awareness among orthodontists, allowing them to produce the most appropriate orthodontic treatment plans for patients.

## 5. Conclusions

In conclusion, the two-generation exposure to HFD affected craniofacial growth and development in a sex-specific manner in young adult rats. It was shown that maternal HFD combined with post-weaning HFD consumption significantly promoted neurocranium and viscerocranium growth in male offspring. In contrast, the mandible and dental morphologies were not affected by the HFD in males or females. These findings demonstrated the intermingled effect of maternal HFD and long-term post-weaning HFD exposure on craniofacial morphology. Nevertheless, more research is needed to test the period-specific (i.e., embryogenesis, pre-, and post-weaning periods) effect of a maternal HFD on offspring. Additionally, the cellular mechanisms of a maternal HFD on offspring’s craniofacial development, as well as therapeutic interventions, should be examined in future studies. 

## Figures and Tables

**Figure 1 jcm-12-01903-f001:**
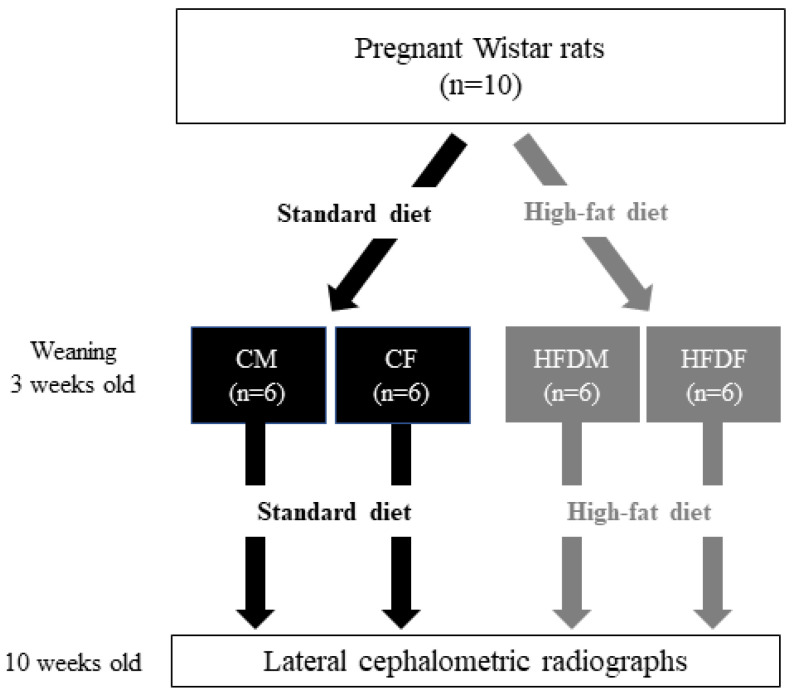
Experimental design. Pregnant rats were fed either a control diet (CE2, Clea, Japan; 4.6% from fat, 3.402 kcal/g) or an HFD (HFD32, Clea, Japan; 32% from fat, 5.076 kcal/g) from day 7 of pregnancy until the end of the lactation period. Twenty-four offspring were selected and assigned to 4 groups: CM (control male, n = 6), CF (control female, n = 6), HFDM (high-fat diet male, n = 6), and HFDF (high-fat diet female, n = 6) groups. Cephalometric radiographs were taken at the age of 10 weeks, just before being sacrificed.

**Figure 2 jcm-12-01903-f002:**
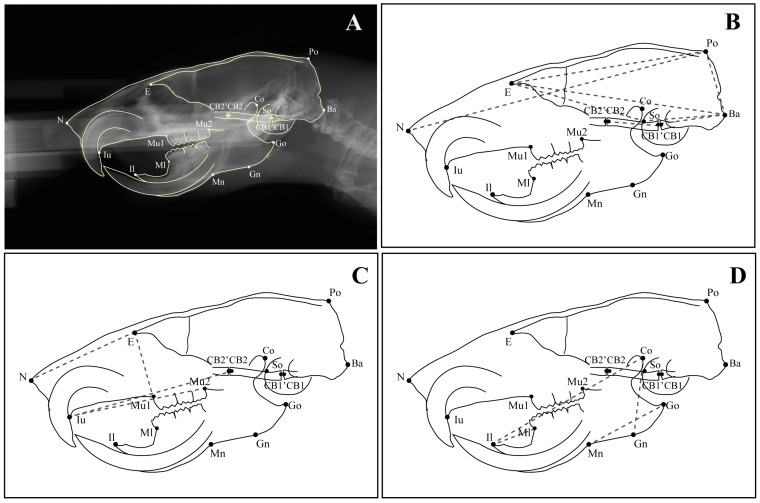
Cephalometric landmarks and linear measurements: (**A**) tracing and cephalometric landmarks on a radiograph; (**B**) linear measurements of the neurocranium consisting of Po–N: total skull length, Po–E: cranial vault length, Ba–E: total cranial base length, So–E: anterior cranial base length, Ba–CB1: occipital bone length, CB10–CB2: sphenoid bone length, Ba–So: posterior cranial base length, and Po–Ba: posterior neurocranium height; (**C**) linear measurements of the viscerocranium consisting of E–N: nasal length, Mu2–Iu: palate length, CB2–Iu: midface length, and E–Mu1: viscerocranial height; (**D**) linear measurements of the mandible consisting of Go–Mn: posterior corpus length, Ml–Il: anterior corpus length, Co–Il: total mandibular length, and Co–Gn: ramus height.

**Figure 3 jcm-12-01903-f003:**
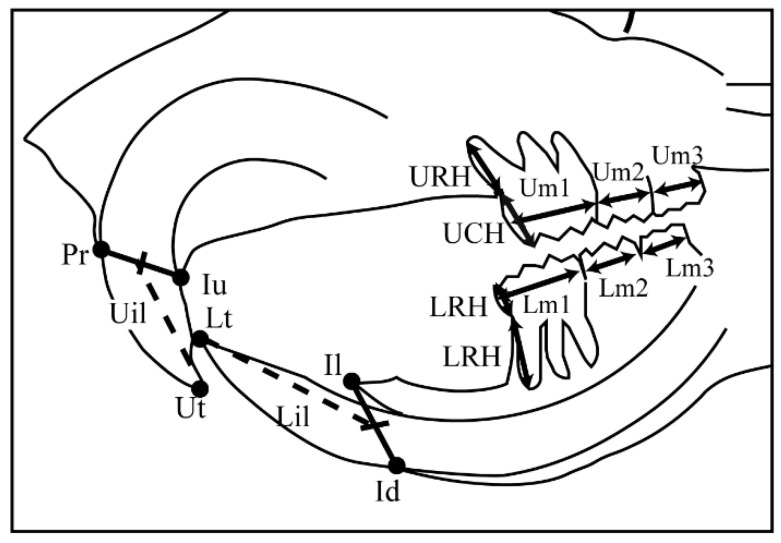
Dental landmarks and linear measurements. All molar crown widths were measured at the widest part of the molar’s dental crown mesiodistally. Um1: maxillary first molar crown width; Um2: maxillary second molar crown width; Um3: maxillary third molar crown width; Lm1: mandibular first molar crown width; Lm2: mandibular second molar crown width; Lm3: mandibular third molar crown width; Uil: maxillary incisor length, measured from the most prominent point between the incisal edges of the upper incisors (Ut) to the middle of the Pr–Iu line; Lil: mandibular incisor length, measured from the most prominent point between the incisal edges of the lower incisors (Lt) to the middle of the Il–Id line; UCH: maxillary first molar crown height, measured from the mesial cusp tip of the first maxillary molar to the cementoenamel junction (CEJ); URH: maxillary first molar mesial root length, measured from the CEF to the mesial root tip of the maxillary first molar; LCH: mandibular first molar crown height, measured from the mesial cusp tip of the first mandibular molar to the CEJ; LRH: mandibular first molar mesial root length, measured from the CEF to the mesial root tip of the mandibular first molar.

**Figure 4 jcm-12-01903-f004:**
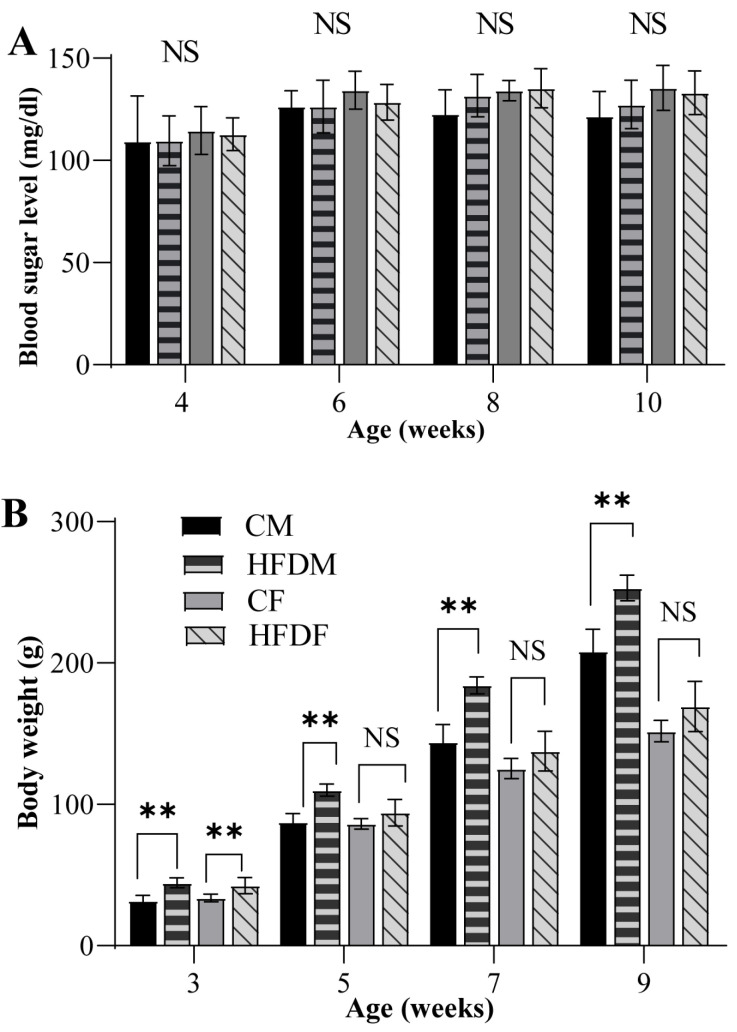
Fasting blood sugar level (**A**) and body weight (**B**) of the CM, HFDM, CF, and HFDF. All data are shown as the mean ± SD (two-way ANOVA). ** *p* ≤ 0.01. NS, not significant.

**Figure 5 jcm-12-01903-f005:**
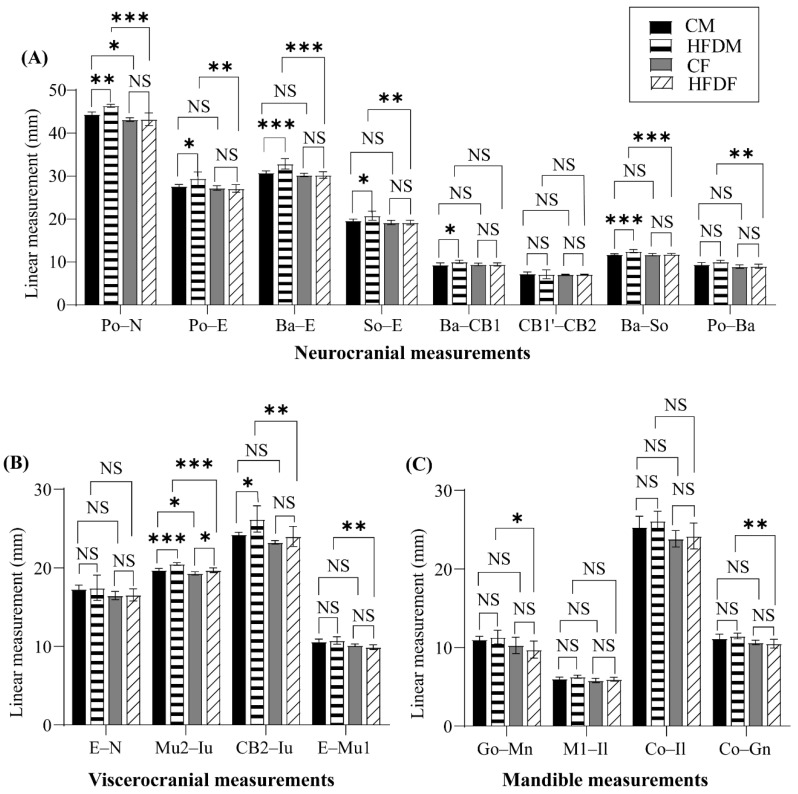
Cephalometric analyses: (**A**) changes in the neurocranial measurements of the CM, HFDM, CF, and HFDF groups; (**B**) changes in the viscerocranial measurements of the CM, HFDM, CF, and HFDF groups; (**C**) changes in the mandible measurements of the CM, HFDM, CF, and HFDF groups. All data are shown as the mean ± SD (two-way ANOVA followed by Tukey’s multiple comparisons test). * *p* < 0.05, ** *p* ≤ 0.01, and *** *p* ≤ 0.0001. NS, not significant.

**Figure 6 jcm-12-01903-f006:**
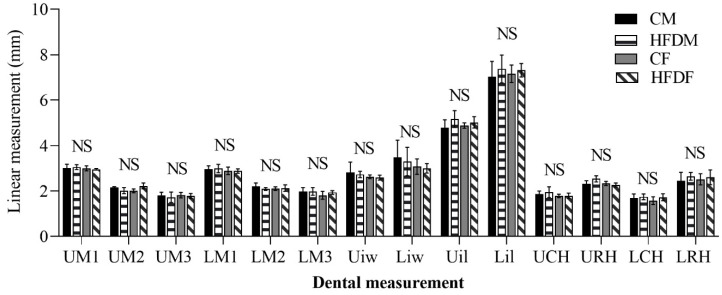
Dental analyses. All data are shown as the mean ± SD (Kruskal–Wallis test). There was no significant difference in any dental measurements. NS, not significant.

**Table 1 jcm-12-01903-t001:** Definitions of radiographic landmarks.

Landmark	Definition
N	The most anterior point on the nasal bone
E	The intersection of the frontal bone and floor of the anterior cranial fossa
Po	The most posterior and superior point on the skull
Ba	The most posterior and inferior point on the occipital condyle
Co	The most posterior and superior point on the mandibular condyle
Go	The most posterior point on the mandibular ramus
Mn	The most concave portion of the concavity on the inferior border of the mandibular corpus
Gn	The most inferior point on the ramus that lies on a perpendicular bisector of the line Go–Mn
I1	The most anterior and superior point on the alveolar bone of the mandibular incisor
So	The intersection of the most anterior tympanic bulla and the superior border of the sphenoid bone
CB1	The most anterior point on the occipital bone at the spheno-occipital synchondrosis
CB1′	The most posterior point on the sphenoid bone at the spheno-occipital synchondrosis
CB2	The most anterior point on the sphenoid bone at the spheno-basispheno synchondrosis
CB2′	The most posterior point on the basisphenoid bone at the spheno-basispheno synchondrosis
M1	The junction of the alveolar bone and the mesial surface of the first mandibular molar
Mu1	The junction of the alveolar bone and the mesial surface of the first maxillary molar
Mu2	The junction of the alveolar bone and the distal surface of the third maxillary molar
Iu	The most anterior and inferior point on the maxilla posterior to the maxillary incisors
Pr	The most anterior and inferior point on the alveolar process of the premaxilla
Id	The most anterior and inferior point on the alveolar process of the mandible
Ut	The most prominent point between the incisal edges of the upper incisors
Lt	The most prominent point between the incisal edges of the lower incisors

**Table 2 jcm-12-01903-t002:** Linear cephalometric variables.

Neurocranium	
Po-N	Total skull length
Po–E	Cranial vault length
Ba–E	Total cranial base length
So–E	Anterior cranial base length
Ba–CB1	Occipital bone length
CB1′–CB2	Sphenoid bone length
Ba–So	Posterior cranial base length
Po–Ba	Posterior neurocranium height
Viscerocranium	
E–N	Nasal length
Mu2–Iu	Palate length
CB2–Iu	Midface length
E–Mu1	Viscerocranial height
Mandible	
Go–Mn	Posterior corpus length
M1–Il	Anterior corpus length
Co–Il	Total mandibular length
Co–Gn	Ramus height

**Table 3 jcm-12-01903-t003:** Linear dental variables.

Dental	
Um1	Maxillary first molar crown width
Um2	Maxillary second molar crown width
Um3	Maxillary third molar crown width
Lm1	Mandibular first molar crown width
Lm2	Mandibular second molar crown width
Lm3	Mandibular third molar width
Uil	Maxillary incisor length
Lil	Mandibular incisor length
UCH	Maxillary first molar crown height
URH	Maxillary first molar mesial root length
LCH	Mandibular first molar crown height
LRH	Mandibular first molar mesial root length

**Table 4 jcm-12-01903-t004:** Main effects and interactions of food and sex on the cephalometric linear parameters.

	Male				Female				Two-Way ANOVA		
	CM		HFDM		CF		HFDF		Effect (*p*-Value)		Interaction
	Mean	SD	Mean	SD	Mean	SD	Mean	SD	Food	Sex	(*p*-Value)
Neurocranium											
Po–N	44.38	0.57	46.40	0.34	43.15	0.43	43.22	1.45	0.015	<0.001	0.024
Po–E	27.65	0.41	29.42	1.55	27.23	0.55	27.12	0.90	0.047	0.002	0.026
Ba–E	30.77	0.40	32.86	1.20	30.26	0.40	30.26	0.77	0.003	<0.001	0.003
So–E	19.66	0.33	20.79	1.07	19.20	0.49	19.17	0.54	NS	0.001	0.047
Ba–CB1	9.35	0.48	10.06	0.37	9.44	0.31	9.44	0.39	0.04	NS	0.037
CB1′–CB2	7.26	0.43	7.15	1.05	7.09	0.08	7.15	0.10	NS	NS	NS
Ba–So	11.78	0.16	12.52	0.35	11.71	0.27	11.79	0.23	<0.001	0.001	0.005
Po–Ba	9.42	0.49	10.05	0.34	8.95	0.42	9.02	0.52	NS	<0.001	NS
Viscerocranium											
E–N	17.31	0.51	17.47	1.61	16.48	0.52	16.57	0.77	NS	0.04	NS
Mu2–Iu	19.73	0.21	20.52	0.17	19.32	0.20	19.71	0.30	<0.001	<0.001	0.042
CB2–Iu	24.27	0.28	26.22	1.67	23.25	0.23	24.01	1.27	0.005	0.001	NS
E–Mu1	10.60	0.34	10.77	0.47	10.15	0.15	9.92	0.30	NS	<0.001	NS
Mandible											
Go–Mn	11.04	0.39	11.33	0.88	10.28	1.03	9.75	1.09	NS	0.004	NS
M1–Il	6.03	0.22	6.29	0.20	5.81	0.28	5.96	0.24	0.042	0.01	NS
Co–Il	25.35	1.35	26.13	1.21	23.86	1.05	24.21	1.65	NS	0.005	NS
Co–Gn	11.17	0.54	11.47	0.37	10.65	0.29	10.50	0.56	NS	<0.001	NS

Two-way ANOVA: NS, not significant; SD, standard deviation.

## Data Availability

Not applicable.

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
