# Peer review of "The Effect of Two-Generation Exposure to a High-Fat Diet on Craniofacial Morphology in Rats"

_jcm, 2023, doi:10.3390/jcm12051903_

Round 1
Reviewer 1 Report
The study by Serirukchutarungsee et al. aims to examine the effects of maternal and postnatal intake of high-fat diets on craniofacial growth in male and female rats. Interestingly, the authors found that male rats fed with high-fat diets showed both increased body weight and larger neurocranial parameters compared to the control male group, whereas there is no significant difference in female rats. Although the study is interesting, there are several major issues. First, although the authors examined four groups (CM,CF, HFDM, and HFDF), all of these groups were sequentially given with same diets. To test the effects separately (and any combinations) during embryogenesis, pre-weaning periods, and post-weaning periods, the authors need to include several additional groups in this study (with switching diets at that timepoints). The quality of image provided in Figure 2 as an example used for cephalometric analysis seems not to be sufficient enough to identify and measure the length and angles between landmarks. How can the authors accurately measure the length less than mm on these images? Another major concern is about blood test. Since the rats were fed with high-fat diets, the authors should test a lipid profile instead of the fasting blood sugar level. Several sentences in the manuscript are overstated without supporting results as pointed out above.
Author Response
We sincerely appreciate Reviewer 1’s valuable time and comments. We have taken due care to revise the manuscript with full consideration of each comment. Please find the attachment for the point-to-point responses.

Reviewer 2 Report
In this manuscript, Ono’s group aims to provide a systemic analysis of craniofacial skeleton morphology based on X-ray results in rats with exposure to a high-fat diet for two generation. This is the first reports showing the effects of two generation of HFD exposure on craniofacial morphology, however, the experimental design is not good enough to get a clear conclusion.
1. Experimental rats exposed to HFD from embryonic stages to 10 weeks old after birth. This long-time exposure will cause misinterpretation of the data. It is difficult to distinguish the differences in HFD-exposed rats are due to maternal exposure or later stage exposure. And also the different effects of HFD on males and females may be different at earlier stages. The authors should increase groups dissected and analyzed at earlier stages, such as embryonic day 21.5 and postnatal day 21 (weaning day).
2. There is no justification for the starting time for HFD treatment in the pregnant rats.
3. There is no justification of the sample number in each group. Is n=6 enough to get trustable results based on statistic analysis? And are these samples from the same litter or different litter? How they assigned into each group? The authors should describe these details clearly in the materials and methods part.
4. The format of the references is not uniform.
5. there are some typos in the manuscript, such as “feq12qmale” in the discussion part.
Author Response
We sincerely appreciate Reviewer 2’s valuable time and comments. We have taken due care to revise the manuscript with full consideration of each comment. Please find the attachment for the point-to-point responses.

Round 2
Reviewer 1 Report
The authors improved the manuscript based on the reviewer's comments.
Reviewer 2 Report
Thanks for the revision. I do not have other comments for this manuscript.